# Predictability of a tornado environment index from ENSO and the Arctic Oscillation

Michael K. Tippett[1], Chiara Lepore[2], and Michelle L. L'Heureux[3]

[1]Department of Applied Physics and Applied Mathematics, Columbia University, New York, New York
[2]Lamont-Doherty Earth Observatory, Columbia University, Palisades, New York
[3]NOAA/NWS/NCEP/Climate Prediction Center, College Park, Maryland

**Correspondence:** M. K. Tippett (mkt14@columbia.edu)

**Abstract.** ENSO modulates severe thunderstorm activity in the U.S., with increased activity expected during La Niña conditions. There is also evidence that severe thunderstorm activity is influenced by the Arctic Oscillation (AO), with the positive phase being associated with enhanced activity. The combined ENSO/AO impact is relevant for situations such as in early 2021 when persistent, strong positive and negative AO events occurred during La Niña conditions. Here we examine the relation of a spatially-resolved tornado environment index (TEI) with ENSO and the AO in climate model forecasts of February, March, and April conditions over North America. Bivariate composites on Niño 3.4 and AO indices show that TEI predictability is high (strong signals and probability shifts) when the ENSO and AO signals reinforce each other and low when they cancel each other. The largest increase in the expected value and variance of TEI occurs when Niño 3.4 is negative and the AO is positive. Signal-to-noise ratios are higher during El Niño/negative AO than during La Niña/positive AO, but probability shifts are comparable.

## 1 Introduction

ENSO modulates severe thunderstorm activity (tornadoes, large hail, and damaging straight-line winds) in the U.S., with increased activity expected during La Niña conditions in winter and spring (Marzban and Schaefer, 2001; Cook and Schaefer, 2008; Allen et al., 2015; Moore, 2019). The association of ENSO with U.S. tornado and hail activity provides a basis for seasonal predictions that are based on observed or predicted values of the Niño 3.4 index (Lepore et al., 2017). On the other hand, La Niña conditions during the beginning of 2021 did not translate into consistently enhanced tornado activity during that period. In particular, Oceanic Niño Index values in 2021 were $-0.9$, $-0.8$, $-0.7$, and $-0.5$ for JFM, FMA, MAM, and AMJ, respectively. However, February, March, April, and May preliminary[1] tornado report numbers from NOAA's Storm Prediction Center were 11, 191, 73, and 289, compared to their most recent three-year averages of 39, 82, 224, and 269, respectively. Only the March preliminary report numbers were above their most recent three-year average, while the February and April preliminary report numbers were below their three-year average. The particular events of early 2021 and the generally modest skill of ENSO-based severe thunderstorm forecasts may simply reflect the limited ENSO signal in severe thunderstorm activity and the large role of unpredictable weather noise, or may indicate the need to consider other factors.

---

[1]Preliminary local storm reports are not quality controlled and are subject to revision.

Climate predictability studies can provide an indication of how much of the observed variability is explained by ENSO and other predictable signals and what skill is to be expected from forecasts. A general challenge in estimating climate predictability is that the predictable variability (signal) is usually modest compared to the unpredictable variability (noise). Predictability can be estimated using observations or physics-based models. An advantage of model-based predictability studies is that sample sizes can be substantially larger than the observational record, and both signal and noise can be better estimated (e.g., Deser et al., 2018). A disadvantage is that the predictability in the model may differ from that in nature (e.g., Scaife and Smith, 2018). Model-based climate predictability studies have tended to focus on quantities such as near-surface temperature and precipitation, and not severe thunderstorm activity, in large part because climate models do not resolve thunderstorms. However, climate models can simulate environmental quantities associated with thunderstorms such as convective available potential energy and wind shear. In the so-called ingredients approach, a climate model simulates the response of these thunderstorm ingredients to prescribed SST (Lee et al., 2012) or radiative forcings (e.g., Lepore et al., 2021, and references therein).

The difference between the expected ENSO response and what occurs in a given year (e.g., in early 2021) can be attributed to atmospheric noise. However, such atmospheric noise may include predictable variability that is independent of ENSO. For instance, ENSO-based predictability of California winter precipitation is relatively low (Kumar and Chen, 2017), but taking account of subseasonal components identifies additional sources of predictability that can explain deviations from the expected ENSO response (L'Heureux et al., 2021; Wang et al., 2017). The Arctic Oscillation (AO) may be such a source of subseasonal predictability for North American severe thunderstorms activity.

The AO is a dominant mode of hemispheric variability which influences North American near-surface temperature and precipitation, especially during the cold season (Thompson and Wallace, 1998). In particular, the positive phase of the AO is associated with warmer temperatures across the eastern U.S. and increased variance of band-passed (3–10 days) 500 haPa height anomalies (Higgins et al., 2000; Thompson and Wallace, 2001). Moreover, the AO is predictable beyond weather time scales, especially in winter (L'Heureux et al., 2017; Riddle et al., 2013; Stockdale et al., 2015; Tang et al., 2007). Origins of AO predictability include the stratosphere (Nie et al., 2019) and the tropics (Kumar and Chen, 2018; Scaife et al., 2017). Some evidence suggests that the AO modulates severe thunderstorms activity. Childs et al. (2018) found a statistically significant relation between the AO and November–February numbers of tornado rated EF1 and higher in the southeastern U.S. Notably those tornado report numbers were more strongly related with the AO than with ENSO. Brown and Nowotarski (2020) reported that daily values of the NOAA/Climate Prediction Center AO index were relevant to tornado outbreak likelihood in the southeastern U.S. across all seasons. Nouri et al. (2021) found a positive relation between annually-averaged state tornado frequencies and the AO. While only suggestive, the preliminary tornado report numbers in early 2021 are not inconsistent with the concurrent values of the AO index: February $-1.191$, March $2.109$, April $-0.204$, and May $-0.161$.

A limitation of previous studies is that they have not considered spatially-resolved, sub-annual, combined impacts of ENSO and AO on severe thunderstorm activity. Pooling data across the U.S. or multi-state regions can mix different climate signal responses, which may dilute or obscure signals and their spatial dependence. Subseasonal analysis is also preferable because of the strong annual cycle in U.S. severe thunderstorm activity and because the persistence of the AO as measured by its auto-correlation function is limited to less than 30 days (Domeisen et al., 2018; Keeley et al., 2009). However, robustly estimating

the spatially-resolved, subseasonal joint response to ENSO and AO from observations alone is challenging because of high sampling variability. Climate models can provide a complementary approach in which sample size is not limited by the length of the observational record.

Here, we examine ENSO and AO signals in monthly climate model forecasts. The number of monthly samples in the climate model data is larger than the observational record by more than a factor of 200 because of multiple forecast initializations and ensemble members. Since the climate model does not resolve thunderstorms, we investigate the predictability of a spatially resolved tornado environment index (TEI). TEI is known to capture some aspects of the observed tornado climatology and variability when it is computed from reanalysis or climate model forecast data (Lepore et al., 2018; Tippett et al., 2014). The work here is an extension of that in Tippett and Lepore (2021), in which we used the same climate model but analyzed the impact of ENSO only on a multi-state average of TEI. Here, we specifically address the following questions: What are the spatially-resolved ENSO and AO signals in TEI? How do the ENSO and AO signals in TEI interfere constructively and destructively? How does the predictability of TEI depend on the joint ENSO/AO phase?

## 2 Data and Methods

### 2.1 Data

The Climate Forecast System, version 2 (CFSv2; Saha et al., 2014) data used is similar to that of Tippett and Lepore (2021) with some modifications that include additional variables and extension of the spatial domain. Monthly values of the Niño 3.4 index, convective precipitation (cPrcp), storm-relative helicity (SRH), and geopotential height at 1000 hPa (Z1000) were taken from reforecasts and realtime forecasts of CFSv2 during the period 1981–2021. Reforecasts consist of 4 initializations per day (0000, 0600, 1200, and 1800) on every fifth day (not counting Feb 29) starting from 12 Dec 1981 and ending on 27 Mar 2011. Realtime forecasts were sampled at the same initialization frequency starting on 2 Apr 2011 and ending on 29 May 2021. Forecast target months that include the initialization date were discarded. Starts and lead times corresponding to February, March, and April monthly targets were used in the predictability analysis (sample sizes of 8,774, 8,498, and 8,537, respectively, for a total of 25,809 forecasts). The Z1000 EOF calculation used 51,653 monthly forecast targets in the range November–May.

Niño 3.4 and Z1000 anomalies were computed with respect to a forecast climatology that is a function of target month and lead time, where lead time is defined as the number of days from the initialization day to the beginning of the target month and ranges from 1 to 276 days ($\sim$9 months). The forecast climatology is computed by averaging over a $\pm$10-day lead-time window. In other words, each forecast anomaly is with respect to the mean of all forecasts that have the same target month and whose lead time is within 10 days of that of the anomaly being computed. Separate Niño 3.4 and Z1000 climatologies were used for starts before and after 0000 1 Jan 1999 to account for a discontinuity in CFSv2 initial conditions (Barnston and Tippett, 2013; Kumar et al., 2012; Xue et al., 2011).

The tornado environment index (TEI) was computed from CFSv2 output on a $1° \times 1°$ latitude-longitude grid for land points in the domain $140°$W to $60°$W and $25°$N to $60°$N according to Lepore et al. (2018)

$$\text{TEI} = \exp(-14.01 + 1.36 \log \text{cPrcp} + 1.89 \log \text{SRH}) \times \text{number of days in target month},$$

where the units of cPrcp and SRH are kg m$^{-2}$ d$^{-1}$ and m$^2$ s$^{-2}$, respectively. The CFSv2 TEI forecast climatology in February has its largest values localized to the Gulf coast which increase and shift northward in March and April (Figure S1).

**2.2    Methods**

We applied EOF analysis to CFSv2 monthly forecasts of hemispheric Z1000 poleward from $20°$N and used PC1 as an AO index (Thompson and Wallace, 1998). EOF1 explained 31% of the total area-weighted monthly variance and is characterized by low-pressure anomalies over the pole and high-pressure anomalies over the midlatitude Pacific and Atlantic basins (Figure S2).

For composites, positive and negative ENSO and AO conditions were defined as occurring when index amplitudes exceeded 0.76 times the monthly standard deviation of the index. We used a lower threshold than the one standard deviation threshold in Tippett and Lepore (2021) to increase the samples sizes of the four possible bivariate (e.g., El Niño *and* positive AO) composites. For a normally distributed random variable, the 0.76 standard deviation threshold corresponds to the upper and lower 22.3% of values. That is, the probability of a normally distributed random variable with mean zero and unit variance

exceeding 0.76 is 22.3%. For the joint occurrence of bivariate independent normally distributed random variables, this threshold corresponds to 5% of values. That is, the probability of two independent and normally distributed random variables with zero mean and unit variance simultaneously exceeding 0.76 is 5% ($= 0.223^2$). Each month has over 8000 samples, and 5% of 8000 is 400. However, the Niño 3.4 and AO indices are negatively correlated in CFSv2 forecasts of February, March, and April (Figure S3). Consequently, the sample sizes of the bivariate composites (corners in Figure S3) differ systemically. In

particular, the composites with opposite-signed Niño 3.4 and AO indices have more samples, and the composites with same-signed Niño 3.4 and AO indices have fewer samples. This difference is largest for March which has the strongest ENSO/AO relation ($r^2 = 13\%$). The fraction of CFSv2 forecast Februaries with negative Niño 3.4 and negative AO indices exceeding the threshold (similar to February 2021) is $262/8774 \approx 3\%$.

     We measured the predictability of TEI during univariate (e.g., El Niño) and bivariate (e.g., El Niño and positive AO) com-

115 posite conditions using skill scores that were computed under the perfect model assumption. No observational data were used. We measured the predictability of deterministic forecasts using the mean squared error skill score (MSESS). The perfect model MSESS is (Tippett and Lepore, 2021)

$$\text{MSESS} = 1 - \frac{\text{MSE}}{\text{MSE}_{\text{clim}}} = \frac{S_c^2}{S_c^2 + \sigma_{X|c}^2} . \tag{1}$$

where $c$ labels the univariate or bivariate condition, $X$ is the variable being predicted, $\text{MSE} = \sigma_{X|c}^2$ is its conditional variance (noise), $\text{MSE}_{\text{clim}} = S_c^2 + \sigma_{X|c}^2$ is the mean squared error of a climatological forecast (bias$^2$ + noise), and $S_c = E[X \mid c] - E[X]$

is the conditional anomaly (signal); $E$ denotes expectation and the vertical line means "conditional on." Perfect model MSESS

varies between zero and one and is an $r^2$-value in the sense that it is the square of the expected value of the anomaly correlation for the given condition (Kumar, 2009; Sardeshmukh et al., 2000). For probability forecasts, we consider the probability of TEI exceeding its monthly climatological median. We measured predictability of probabilistic forecasts in terms of the shift of the forecast probability away from its climatological value of 0.5 because the perfect model values of the Brier and log skill scores depend only on the size of the probability shift—larger probability shifts result in larger skill scores (Tippett and Lepore, 2021). Expected skill scores in the perfect model context here are always positive (more skill than climatology).

We computed the empirical (no fitting) cumulative probability distribution function of the area-weighted sum of TEI over land points east of $110°$W (denoted summed TEI) for different conditions. We plotted in the results in form of return level plots in which the vertical coordinate is the $100 \times (1 - p)$ percentile of the data, the horizontal coordinate is the approximate return period $y_p = -1/\log(1 - p) \approx 1/p$, and $p$ is a probability (Coles, 2001; DelSole and Tippett, 2022). For instance, $p = 0.01$ corresponds to the 100-year return level and period. For $p = 0.5$, the return level is the median, $y_{0.5} \approx 1.44$ and is marked on the return level plots with M.

To assess the statistical significance of composite, correlation, and probability maps, we followed the procedure of Benjamini and Hochberg (1995) as detailed in Section 13.4 of DelSole and Tippett (2022). First, a two-sided p-value is computed at each grid point (land only). The p-value for the correlation $\hat{\rho}$ and sample size $N$ is computed by considering the quantity

$$t_{\text{correlation}} = \frac{\hat{\rho}\sqrt{N - 1}}{\sqrt{1 - \hat{\rho}}},$$

which has a t-distribution with $N - 2$ degrees of freedom under the null hypothesis of no correlation. The p-value of a composite under the condition $c$ is computed by considering the quantity

$$t_{\text{composite}} = \frac{\hat{\mu}_c - \hat{\mu}_{\overline{c}}}{\hat{\sigma}_{\text{pooled}}\sqrt{\frac{1}{N_c} + \frac{1}{N_{\overline{c}}}}}$$

which has a t-distribution with $N_c + N_{\overline{c}} - 2$ degrees of freedom under the null hypothesis of no difference; $\hat{\mu}$ and $N$ are the sample mean and sample size, respectively, under the condition $c$ and its negation $\overline{c}$, as indicated by the subscript, and the pooled correlation is

$$\hat{\sigma}_{\text{pooled}}^2 = \frac{(N_c - 1)\hat{\sigma}_c^2 + (N_{\overline{c}} - 1)\hat{\sigma}_{\overline{c}}^2}{N_c + N_{\overline{c}} - 2}.$$

The p-value for the probability $\hat{P}_c$ of exceeding the median under condition $c$ is computed from the binomial distribution with $N_c$ trials and success probability 0.5. Second, the p-values are sorted from smallest to largest and then compared to the sequence $\gamma/S, 2\gamma/S, \ldots, \gamma$, where $S$ is the number of land grid points (here $S = 1740$), and $\gamma$ is the specified False Discovery Rate (FDR), here 5%. The null hypothesis is rejected for those p-values that are smaller than the comparison sequence. The largest correlation in absolute value for which the null hypothesis is accepted is denoted $\rho_{\text{FDR}}$; all statistically insignificant correlations have amplitude less than $\rho_{\text{FDR}}$. The largest TEI composite for which the null hypothesis is accepted is denoted $\text{TEI}_{\text{FDR}}$; all statistically insignificant TEI composites have amplitude less than $\text{TEI}_{\text{FDR}}$. The largest probability shift from 50% for which the null hypothesis is accepted is denoted $\text{P}_{\text{FDR}}$; all statistically insignificant probability shifts are less than $\text{P}_{\text{FDR}}$.

Statistical significance of regression maps is equivalent to that of correlation maps (see Section 9.9 DelSole and Tippett, 2022). Statistical significance of a MSESS map under the condition $c$ is equivalent to statistical significance of a composite map under the same condition since analysis of variance in this case is equivalent to a t-test for a difference in means.

For plotting composite, MSESS, and probability maps, we masked locations where the values were statistically insignificant or where the absolute value of the TEI composite anomaly was less than 0.05. In addition, we masked MSESS values less than 0.05 and probability values that were less than 5 percentage points away from 50%. Our use of thresholds in addition to statistical significance reflects the fact that with large sample sizes nearly all results are statistically significant.

## 3   Results

### 3.1   Univariate composites

ENSO and AO composites of TEI anomalies show signals that are centered over Louisiana and Arkansas in February and that shift northward in March and April (Figure 1). The TEI signals are mostly positive during La Niña conditions and during the positive phase of the AO, with the opposite sign over Florida. TEI signals are essentially reversed when the ENSO or AO phase is inverted. Overall, there is no strong indication of nonlinear responses to positive and negative values of the indices.

Signal amplitudes are highest in March and lowest in April. Signals in May are weaker still (not shown). Regression and correlation maps show the same subseasonal variation in the strength of the relation of TEI with the Niño 3.4 and AO indices (Figures S4 and S5). Correlation maps show additional continental scale structure in the west and north where there are sizable correlations but where the TEI variability is too small to appear in the composite or regression patterns. The strongest TEI correlations are of the order 0.3–0.4 which is highly statistically significant in the model data here, but would be less so in 40 years of data for which the 5% significance threshold would be about 0.32. The ENSO and AO spatial patterns and amplitude are similar in February and March. In April the AO pattern is shifted further northward than the ENSO one. Correlation maps indicate that both TEI ingredients contribute to the April differences between ENSO and AO patterns (Figures S6 and S7). Overall, the ENSO signal is slightly stronger than the AO one (Figure 1), which may reflect the stronger correlation of the dominant TEI ingredient (SRH) with the Niño 3.4 index than with the AO index.

To address the question of how ENSO and AO might modulate the total number of tornadoes, we examined the distribution of TEI summed over land points east of $110°$W (last row Figure 1). Summed TEI return levels are higher for La Niña and positive AO (active phases) than for El Niño and negative AO (inactive phases), with corresponding changes in return period. A summed TEI value of 150 in February has a return period of about 10 years during active phases and a return period of about 100 years during inactive phases. Return levels for the two active phases are similar for return periods up to about 20 years at which point sampling variability becomes noticeable, and likewise for the two inactive phases. The active phase return level curves have steeper slopes, and the steeper slopes indicate greater extension of the distribution rightward to more extreme values. Moreover, the return level curves are approximately straight lines, which means that the distributions are reasonably approximated by the Gumbel distribution. The variance of a Gumbel distributed random variable is proportional to the square of the slope of the return level line (see Appendix A) which means that the active phases have higher variance than the inactive

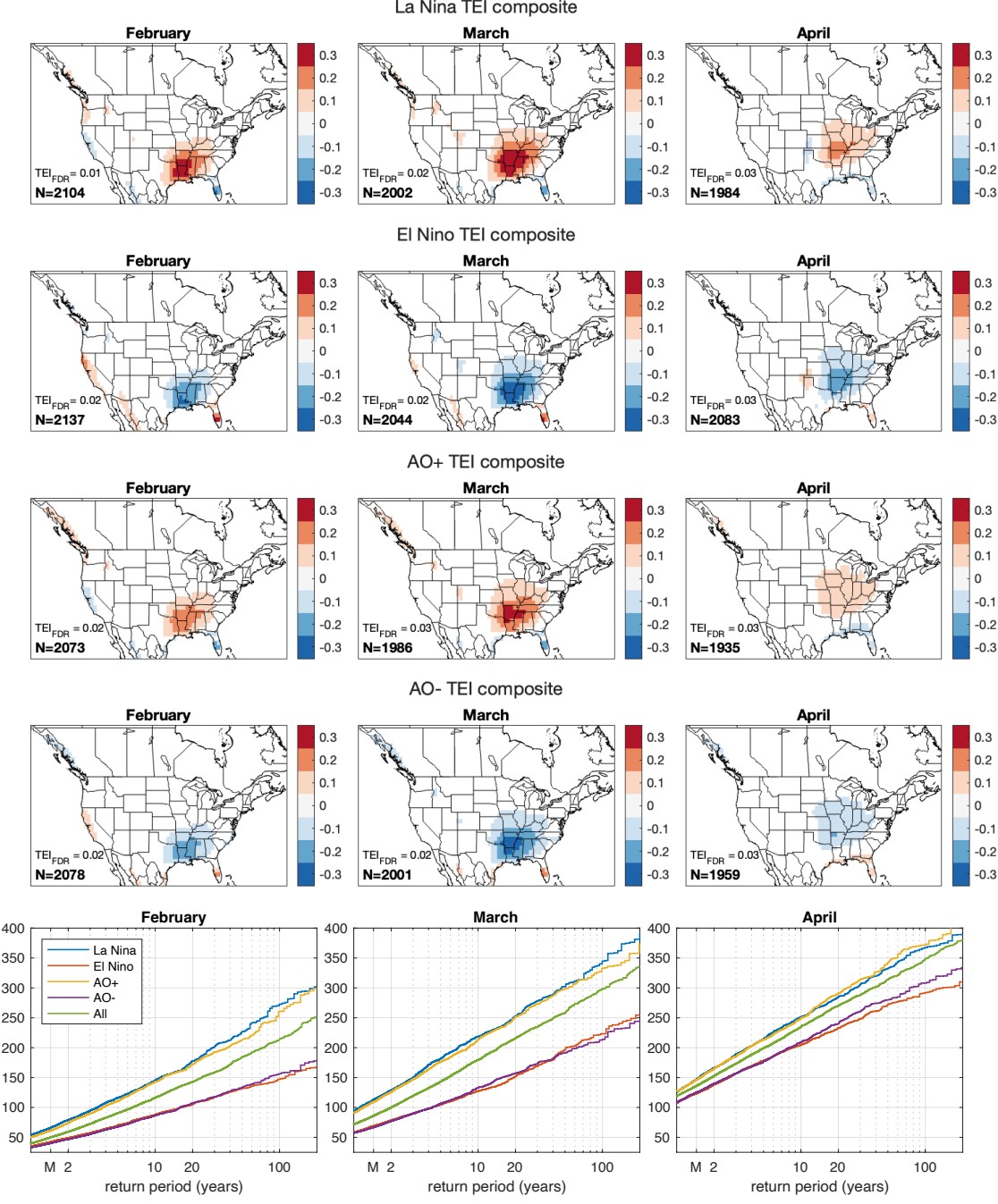

**Figure 1.** Rows 1–4: Univariate Niño 3.4 and AO composites of February, March, and April TEI anomalies. TEI units are number of tornado reports per 1x1 grid box. The sample size for each case is indicated in the lower left corner. All statistically insignificant values have amplitude less than $TEI_{FDR}$ which is shown on each map. Statistically insignificant values and values with absolute value less than 0.05 are masked. Bottom row: Summed (land points east of $110°W$) TEI return levels and approximate return periods conditional on univariate Niño 3.4 and AO phases. 'M' marks the approximate return period of the median.

phases. The differing slopes of the return level curves means that the return level change between active and inactive phases (vertical distance between return levels curves) increases with return period.

## 3.2 Bivariate composites

We computed bivariate composites of TEI anomalies conditional on the simultaneous values of the Niño 3.4 and AO indices to investigate the constructive and destructive interference between the ENSO and AO signals. TEI signals are strong when
the ENSO and AO signals reinforce each other (interference is constructive), which is the case for opposite-signed indices, namely, La Niña/AO+ (first row Figure 2) and El Niño/AO− (fourth row). The bivariate constructive signals are stronger than the univariate ones (compare with Figure 1) and show the same subseasonal variation in strength with the strongest signals in March and the weakest ones in April. TEI signals are weak when the ENSO and AO signals cancel, i.e., same-signed indices— La Niña/AO− and El Niño/AO+ (second and third rows of Figure 2). The La Niña/AO− signal, although weak, is overall
positive.

Summed TEI return levels deviate from their climatological (All) values only when the ENSO and AO signals reinforce each other (bottom row Figure 2). When the ENSO and AO signal cancel, the distribution of summed TEI values is similar to the climatological one. As in the univariate composites, the return level curves are approximately straight lines, and the distributions are reasonably approximated by Gumbel distributions. The active phase return period curves have steeper slopes
than the inactive phase ones, which as in the univariate composites indicate greater extension of the distribution rightward to more extreme values, higher variance, and larger differences at longer return periods. A March summed TEI value of 150 has an approximate return period of 3 years during La Niña/AO+ conditions and a return period of 20 years during El Niño/AO− conditions.

## 3.3 Bivariate composite predictability

Perfect model MSESS values can be interpreted as squared anomaly correlation values and are low when the ENSO and AO signals cancel (second and third rows of Figure 3). Comparing the two constructively phased cases, MSESS is higher when TEI is reduced (El Niño/AO−) than when TEI is enhanced (La Niña/AO+). Since MSESS is an increasing function of signal-to-noise ratio (see Equation 1) and the signal amplitudes of the two constructive phased cases are about the same (compare first and fourth rows of Figure 2), the difference in MSESS is due to the noise variance being larger when TEI is larger. We return
to the reason for this difference in variance in the Discussion. This increased variance is consistent with the increased variance of summed TEI seen in the return level plots (bottom row Figure 2). A consequence of increased variance is that mean squared error will be larger during La Niña/AO+ conditions than during El Niño/AO− conditions.

For probabilistic forecasts during each of the four bivariate conditions, we considered the probability of TEI exceeding its climatological median. The expected Brier skill score and log skill score are symmetric increasing functions of the fore-
215 cast probability shift away from its climatological value of 50%—larger probability shifts mean larger expected skill scores. Therefore, we only show the probability shifts for the bivariate composites (Figure 4). Substantial probability shifts only occur when the ENSO and AO signals reinforce each other, i.e., for bivariate composites with opposite-signed indices, namely La

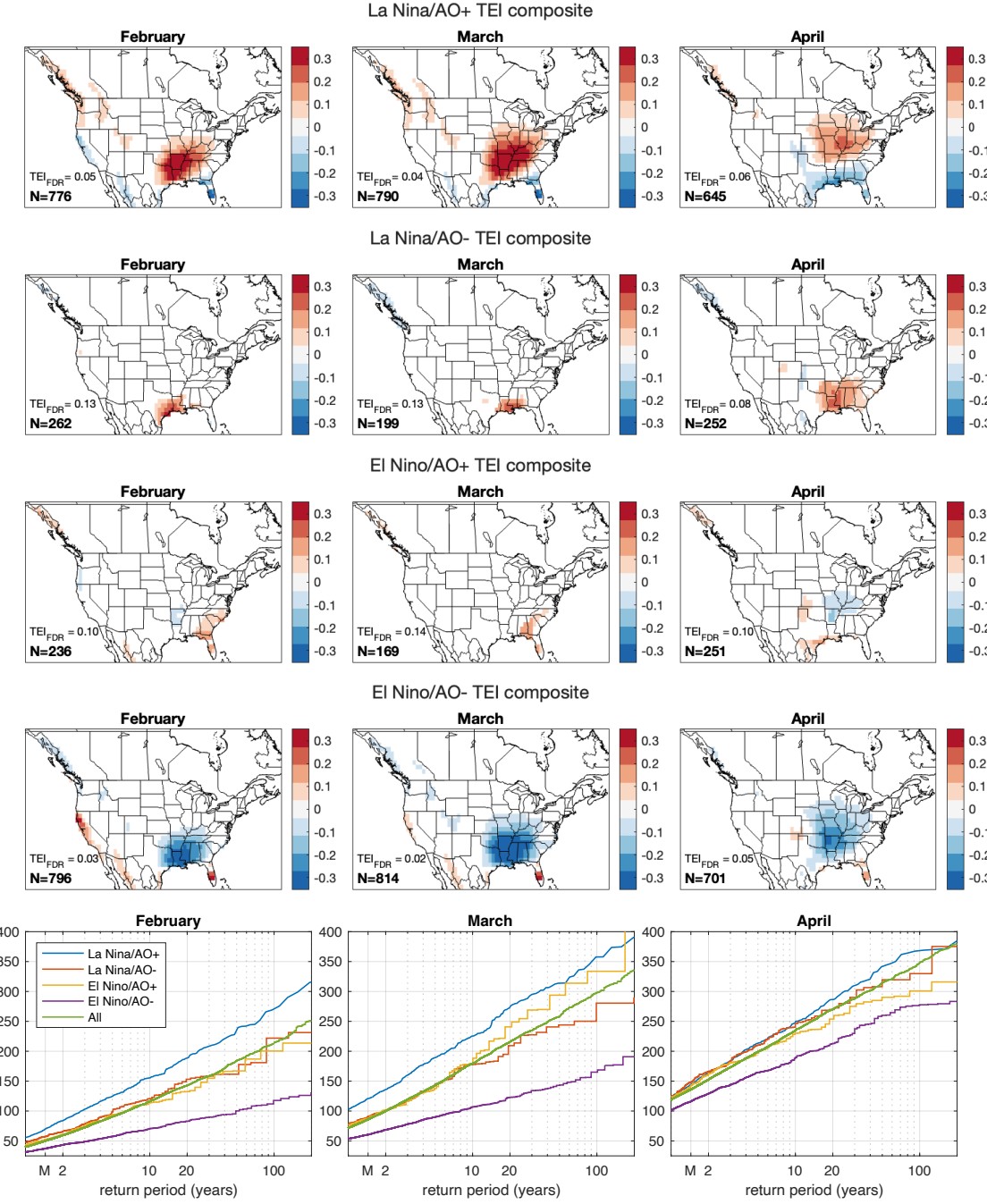

**Figure 2.** Rows 1–4: Bivariate Niño 3.4/AO composites of February, March, and April TEI anomalies. TEI units are number of tornado reports per 1x1 grid box. The sample size for each case is indicated in the lower left corner. All statistically insignificant values have amplitude less than TEI$_{FDR}$ which is shown on each map. Statistically insignificant values and values with absolute value less than 0.05 are masked. Bottom row: Summed (land points east of 110°W) TEI return levels and approximate return periods conditional on bivariate Niño 3.4/AO phase. 'M' marks the approximate return period of the median.

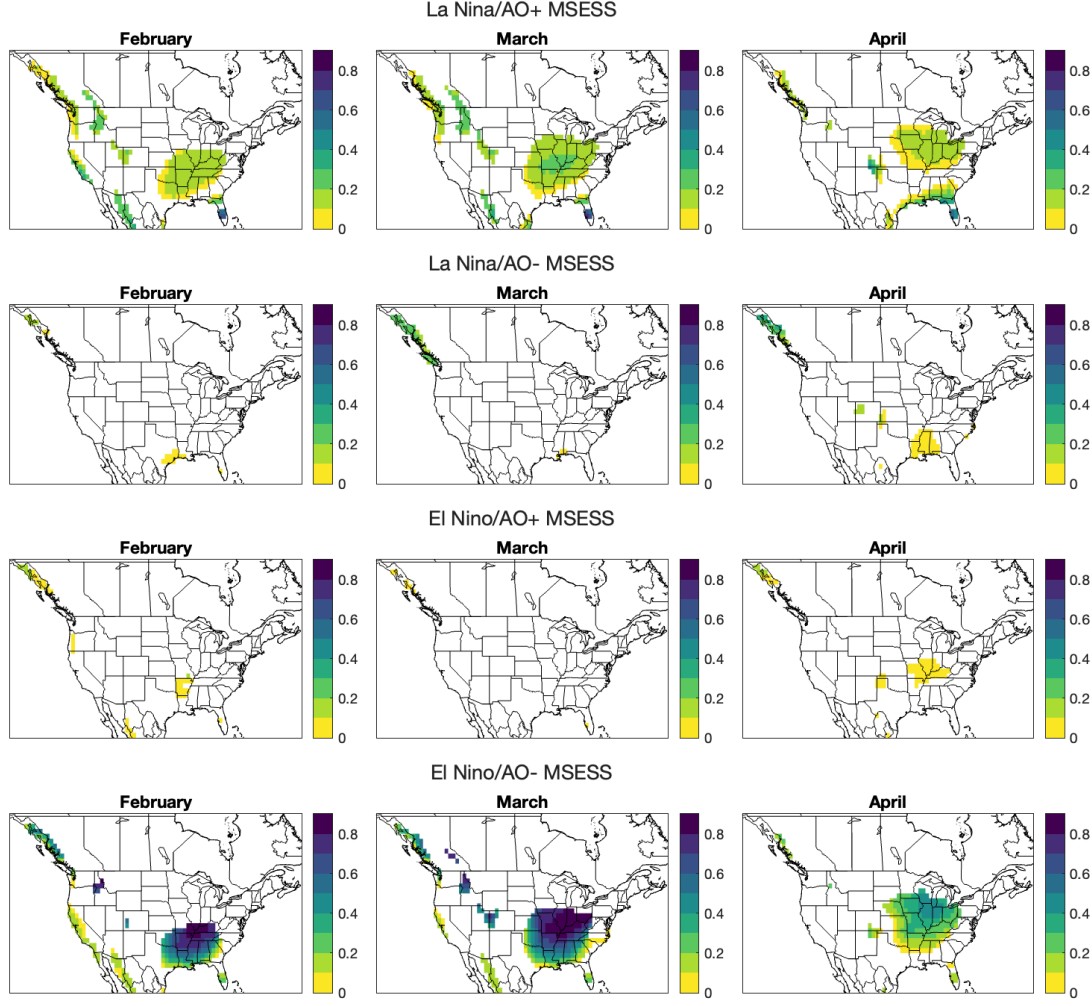

**Figure 3.** Perfect model mean squared error skill score (MSESS) for February, March, and April bivariate Niño 3.4/AO composites of TEI. MSESS values are masked as in Figure 2 and where MSESS values are less than 0.05.

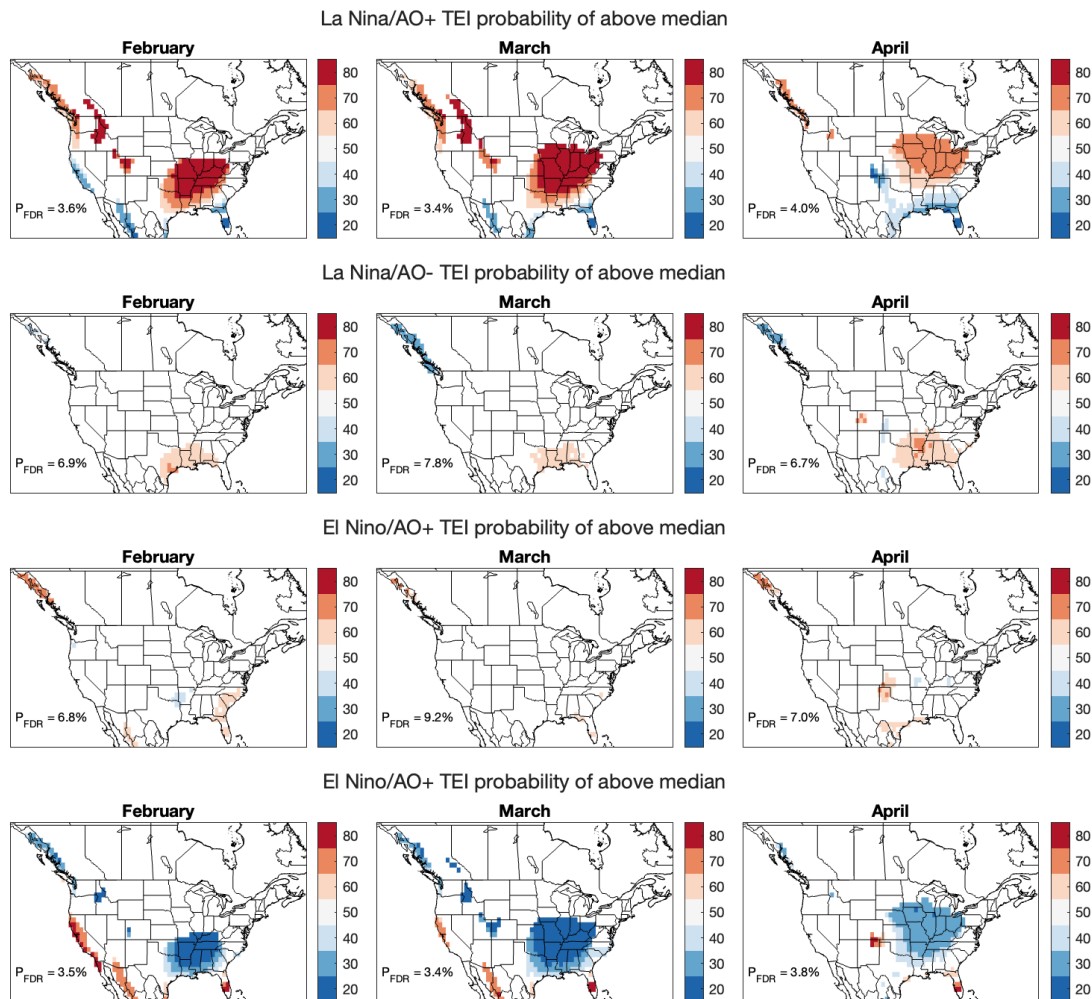

**Figure 4.** Probabilities of February, March, and April TEI exceeding its climatological median value for bivariate Niño 3.4/AO composites. The color bar is centered on the climatological value of 50%. Statistically insignificant values, locations with composite amplitude less than 0.05, and shifts away from 50% that are less than 5 percentage points are masked. All statistically insignificant probability shifts away from 50% are less than $P_{FDR}$ which is shown on each map.

Niña/AO+ and El Niño/AO−. The probability shifts for these two case are nearly the same, indicating the same level of predictability and expected perfect model skill. This behavior is different from that of MSESS which showed higher predictability and expected skill for El Niño/AO−. We return to this point in the Discussion section. The largest probability shifts occur in March, and the smallest probability shifts occur in April. Probability shifts when the ENSO or AO phase is considered separately are weaker than those when ENSO and the AO reinforce each other (compare Figure S8 with the first and fourth rows of Figure 4).

Both predictability measures show some regions west of 100°W and north into Idaho, Washington state, and British Columbia where TEI in CFSv2 is predictable. On the other hand, few if any tornadoes or thunderstorms are reported in some of these regions, and the findings here may reflect a previously noted positive bias of TEI compared to tornado reports in environments that have relatively high SRH and low cPrcp during this time of the year (Tippett et al., 2014).

## 4   Summary and discussion

Preliminary reports of U.S. tornadoes appear to have diverged from the enhanced activity that would be expected during the La Niña conditions of early 2021, a period when notable monthly Arctic Oscillation (AO) anomalies also occurred. To investigate the question of how ENSO and the AO jointly modulate North America severe thunderstorm activity, we computed a tornado environment index (TEI) in 41 years of climate model forecasts for target months in the range February–April. Because the forecasts have many initializations and ensemble members, the sample size is large enough to compute robust bivariate composites based on simultaneous values of the Niño 3.4 and AO indices. Because lead times extend up to about nine months when forecasts are nearly independent of the verifying observations, model results are less closely tied to the observational record of the particular weather events that occurred. Our main findings are:

- ENSO and AO teleconnections in TEI have similar patterns and amplitudes over North America, with the AO index being overall positively correlated with TEI.

- TEI predictability is high (strong anomalies and probability shifts) when the ENSO and AO signals reinforce each other (opposite-signed Niño 3.4 and AO indices).

- When the ENSO and AO signals interfere destructively (same-signed Niño 3.4 and AO indices), the signals cancel, and TEI predictability is small.

We computed the predictability of TEI by target month conditional on the simultaneous phases of ENSO and the AO. Predictability was measured using skill scores that were computed under the perfect model assumption. The mean squared error skill score (MSESS) is a skill score for deterministic forecasts, and the perfect model MSESS depends only on the signal-to-noise ratio. To leading order, MSESS reflects the TEI signal amplitude and is small (little predictability) when the ENSO and AO signals cancel and is large (high predictability) when they reinforce each other. MSESS is highest in March and lowest in April. Comparing the two constructively phased situations, MSESS is higher during inactive phases (positive Niño 3.4 and negative AO indices) than during active phases (negative Niño 3.4 and positive AO indices). The reason for this difference is that the noise variance is smaller during inactive phases, and consequently the signal-to-noise-ratio is larger.

On the other hand, the perfect model Brier and log skill scores depend only on the size of the probability shifts, which are nearly the same for active and inactive constructively phased composites. This difference between predictability as measured by MSESS and predictability as measured by probability shift is perhaps unexpected because previous studies have noted a one-to-one correspondence between perfect model skill scores of deterministic and probabilistic forecasts (e.g., Tippett, 2019, and references therein). For instance, Tippett et al. (2010) found that the perfect model Brier skill score (BSS) was a

function of the anomaly correlation AC and that $BSS \approx 1 - \sqrt{1 - AC}$. However, those results apply when the skill score is computed by averaging over joint-Gaussian distributed forecasts and observations. Here the distributions are not Gaussian, and the averages are over composites which each have a specific mean (signal) and variance (noise). In this case, MSESS averaged over composites is a function of the mean-to-variance ratio alone, but the probability shift is not, even for Gaussian distributions (Tippett et al., 2010). Here areal sums of TEI are Gumbel distributed, and the probability shift for Gumbel distributions depends on the location and scale parameters separately (see Appendix A). Arguably, probability shifts are the more valid predictability measure in this context since they measure the difference between forecast and climatological distributions and since the log skill score is an information theory-based measure (DelSole and Tippett, 2007).

TEI is the product of convective precipitation and storm relative helicity (SRH), and here both factors are sensitive to the phases of ENSO and the AO, with SRH showing stronger correlations (Figures S6 and S7). Tippett and Lepore (2021) showed that the variance of a product of random variables is larger when the means of the two factors are larger and smaller when the means are smaller, which explains the decreased TEI variance during inactive phases seen here. The dependence of both TEI ingredients on ENSO and AO phase is different from the projected climate change signal in which a warming climate leads to upward trends in convective available potential energy (CAPE) and little or downward trends in SRH or other measures of wind shear (Diffenbaugh et al., 2013; Lepore et al., 2021). This difference in dependence might be useful in distinguishing between climate change and internal variability in observations, especially since some observational studies that have detected trends in thunderstorm report data have also found trends in SRH to be the dominant factor (Lu et al., 2015; Tippett, 2014; Tippett et al., 2016). Given the relation between Pacific forcing and SRH seen here, observed trends in the Pacific zonal SST gradient toward a more La Niña-like state might play a role in observed upward SRH trends, though at present whether the Pacific trends represent forced or internal variability is a topic of debate (Seager et al., 2019; Watanabe et al., 2021). The presence of ENSO and AO signals in SRH may also have implications for changes in intensity. Lepore and Tippett (2020) found that increases in SRH were associated with larger percent increases in the number of tornadoes rated EF2 and higher than in lower rated tornadoes. Here the implication would be that the ENSO and AO phases might modulate the relative frequency of stronger tornadoes while the projected climate change signal would not.

Although the model results here suggest a potential role for the joint phases of ENSO and the AO in modulating severe thunderstorm activity, a number of questions remain. Two key questions are whether the ENSO and AO teleconnections in TEI found here are present in other climate models and in reanalysis and whether relations with TEI translate to relations with severe thunderstorm reports. These questions have been explored for the ENSO signal (Allen et al., 2015) but not for the AO and not for ENSO and the AO jointly. As far as we know, this is the first study to examine the constructive/destructive interference of the ENSO and AO signals. Interference of the ENSO and AO signals may also be present in near-surface temperature and precipitation. Regarding the physical mechanisms behind this interaction, one clue might be the fact that the midlatitude jet stream tends to be farther north during both La Niña and positive AO conditions.

Sampling variability is a challenge to analyzing climate signals in severe thunderstorm reports and reanalysis data. The teleconnection patterns found here could provide guidance when pooling observational data in time and space so as to reduce noise without diluting the signal. For instance, the modest signals in April and May would suggest that pooling data across the

March–May season would be suboptimal. In the same vein, analysis of observational data for evidence of an AO signal may be more effective using daily data because the persistence of the AO as measured by its autocorrelation function tends to be less than 30 days (Keeley et al., 2009; Domeisen et al., 2018).

## Appendix A: Gumbel distribution

The CDF of a Gumbel-distributed random variable $X$ is

$$\text{Prob}(X < x) = F(x, \mu, \beta) = e^{-e^{-(x-\mu)/\beta}}$$

where $\mu$ and $\beta > 0$ are location and scale parameters, respectively. The variance of $X$ is $\pi^2 \beta^2/6$. Defining the exceedence probability $p = \text{Prob}(X > x)$, and solving for $x$ gives

$$p = 1 - e^{-e^{-(x-\mu)/\beta}}$$
$$\log(1-p) = -e^{-(x-\mu)/\beta}$$
$$\log(-\log(1-p)) = -\frac{x-\mu}{\beta}$$
$$x = \mu + \beta \log(-1/\log(1-p)).$$

This means that the graph of the return level as function of the approximate return period $y_p = -1/\log(1-p)$ is a straight line on an abscissa log-scale plot. The slope of the line is $\beta$, and the intercept is $\mu$. The median $x_M$ is found by setting $p = 0.5$,

$$x_M = \mu - \beta \log(\log(2)).$$

The approximate return period for the median is $y_{0.5} = 1/\log(2) \approx 1.44$. The mean is $\mu + \beta\gamma$ where $\gamma$ is Euler's constant $\approx 0.577$. Since $-\log(\log(2)) \approx 0.37$, the mean is to the right of median.

Forecasts are of the probability of TEI exceeding its median value conditional on the phases of ENSO and the AO. Suppose that during a particular phase of ENSO and the AO, the Gumbel parameters of the TEI distribution are $\mu + \Delta\mu$ and $\beta + \Delta\beta$, where $\mu$ and $\beta$ are the parameters of the climatological distribution. How does the probability of TEI exceeding its median value change from its climatological value of 50%?

$$P(X > x_M) = 1 - F(x_m, \mu + \Delta\mu, \beta + \Delta\beta)$$

The power series approximation of $F(x_m, \mu + \Delta\mu, \beta + \Delta\beta)$ is

$$F(x_m, \mu + \Delta\mu, \beta + \Delta\beta) \approx F(x_M) + \frac{\partial F}{\partial \mu}\Delta\mu + \frac{\partial F}{\partial \beta}\Delta\beta$$
$$= F(x_M)\left(1 + \log 2\left(\frac{\Delta\mu}{\beta} - \log(\log(2))\frac{\Delta\beta}{\beta}\right)\right). \tag{A1}$$

This means that positive values of $\Delta\mu$ and $\Delta\beta$ increase the probability and negative values decrease the probability.

*Data availability.* ONI data provided by NOAA/CPC at https://origin.cpc.ncep.noaa.gov/products/analysis_monitoring/ensostuff/ONI_v5.
php. AO index data provided by NOAA/CPC at https://www.cpc.ncep.noaa.gov/products/precip/CWlink/daily_ao_index/monthly.ao.index.

b50.current.ascii.table. Preliminary US 2021 tornado report numbers and 3-year averages provided by NOAA/SPC at https://www.spc.noaa.
gov/climo/online/monthly/newm.html. CFSv2 data provided by the IRI Data Library at http://iridl.ldeo.columbia.edu/SOURCES/.NOAA/
.NCEP/.EMC/.CFSv2/.

*Author contributions.* MKT carried out the analysis and prepared the manuscript with contributions from all co-authors.

*Competing interests.* The authors declare no competing interests.

*Acknowledgements.* MKT and CL gratefully acknowledge the support of the Willis Research Network.

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
