# Peer review of "Predictability of a tornado environment index from ENSO and the Arctic Oscillation"

_Weather and Climate Dynamics, 2022_

## Author Comment (AC1)

Responses (in blue) to reviewer comments (in black)

Review 1.

This is a well-put-together manuscript on a timely and important topic. The science builds on previous work and provides results that contribute to our collective understanding of seasonal and sub-seasonal tornado outlooks.

Thanks for the encouraging words and detailed reading.

I have only minor comments for consideration:

Line 20: remove the second "reports" (after "numbers")

Done.

Line 21-23: It is pointed out that the limited predictability of severe thunderstorm activity with ENSO along with unpredictable weather noise likely explain the early-2021 inactive period, despite La Nina conditions being present. I think the authors should add something explicitly about the need to consider multiple oscillations (like AO) to explain more variability. This is hinted at with "limited predictability of ENSO" and especially in line 34, but (again) I think it would set the stage for this work if added here.

We agree and tried to bring in the idea of looking at other predictable signals. We may revise further. Original text: "The particular events of early 2021 and the generally modest skill of ENSO-based severe thunderstorm forecasts may simply reflect the limited predictability of severe thunderstorm activity from ENSO state and the large role of unpredictable weather noise, neither of which is known with great certainty."

Revised text: "The particular events of early 2021 and the generally modest skill of ENSO-based severe thunderstorm forecasts may simply reflect the limited ENSO signal in severe thunderstorm activity and the large role of unpredictable weather noise, neither of which is known with great certainty.

Climate predictability studies can provide an indication of how much of the observed variability is explained by ENSO **and other predictable signals** and what skill is to be expected from forecasts."

Line 43: hyphenate "time-scales"

After checking, "timescales" seems preferable here. (We had two words originally.)

Line 44: I don't understand the point of the sentence beginning "Origins of AO skill include..."

We are reminding the reader why the AO is predictable beyond the timescale of weather. If the AO were not predictable in advance, its relation with storm activity would be of limited predictive utility. We have changed "skill" to predictability to make a connection with the previous sentence.

Line 115: Change "was" to "were" (after "data")

Done

Figure 1: I suggest using the same scale for all maps in rows 1-4. Likewise, use the same scale for all plots in row 5.

Done.

Figure 2: I suggest to use the same scale for all plots in row 5.

Done.

Line 201: Change "Americal" to "American"

Done.

Line 226: Change "to the case" to "when"

Done.

Lines: 237-246: Not a comment for improvement, just want to say that these comments are insightful, and should spur additional studies as we continue to try to understand trends in severe weather ingredients---I've wondered for a while why numerous observational studies have captured increasing trends in SRH (despite the thought that shear-related metric should decline as the planet warms).

Your comment motivated us to mention the current debate about whether observed Pacific SST trends are forced or not, which have implications for the future in this context.

"Given the relation between Pacific forcing and SRH seen here, observed trends in the Pacific zonal gradient toward a more La Niña-like state might play a role in observed upward SRH trends, though at present whether the Pacific trends represent forced or internal variability is a topic of debate (Seager et al., 2019, Watanabe et al., 2021)."

Review 2.

Building incrementally on the author's previous work, this manuscript examines ENSO and AO signals in monthly climate forecasts to examine variability in a derived tornado environment index. The manuscript is reads well and the topical area is suitable for publication in WCD. I have some major concerns about several aspects of the paper, including availability of the tornado environment index data, that are elaborated on below and should be addressed prior to undergoing additional peer-review.

In general, this work offers incremental improvement to existing knowledge. The authors state this incremental advancement. The figures are of good quality, but the lack of representation of statistical significance should be corrected.

**Major Comments:**
1. The title would suggest that the authors examined predictability of the TEI from ENSO and AO. Yet, this analysis was performed using CFSv2, and thus, is bound by the predictability of parent model and has little to do with the representation of arbitrary teleconnection patterns calculated by the authors. In fact, the authors have already published on the skill of weekly and monthly forecasts of the TEI from CFSv2 forecasts. To me, simply examining the teleconnection patterns in the model forecasts is not novel enough to warrant publication, especially considering the rather diluted discussion and findings presented in the results.

We see no evidence that the teleconnection patterns that we have computed are arbitrary. ENSO is characterized using the standard Nino 3.4 index and the AO by the first EOF of Z1000.

Predictability studies are different from skill assessments. (See for instance Section 17.8 What is the Difference Between Predictability and Skill? in DelSole and Tippett 2022). Our previous skill assessment of monthly TEI forecasts from CFSv2 does not provide direct information about the ENSO and AO signals. We do not recall a skill assessment of weekly TEI forecasts from CFSv2, but may have overlooked something.

Our use of the term predictability to describe such model-based results is standard. There is a long history in the scientific literature of predictability studies which are based on model experiments where ensembles provide estimates of the distribution of outcomes conditional on an initial state or climate forcing (e.g., Shukla, 1998). In particular, initialized and uninitialized model-based studies have been used to estimate the range of ENSO impacts (Deser et al., 2018) and rainfall extremes (Thompson et al, 2017).

As we state in the introduction, previous predictability studies have tended to focus on near-surface temperature and precipitation. The predictability study here is one of the few to look at teleconnections of quantities related to severe thunderstorms in a model and the first, as far as we know, to look at AO teleconnections in this way.

DelSole, T. and Tippett, M. K.: Statistical Methods for Climate Scientists, Cambridge University Press, 2022, https://doi.org/10.1017/9781108659055

Shukla, J., 1998. Predictability in the midst of chaos: A scientific basis for climate forecasting. *Science*, *282*(5389), pp.728-731.

Deser, C., Simpson, I. R., Phillips, A. S., & McKinnon, K. A. (2018). How Well Do We Know ENSO's Climate Impacts over North America, and How Do We Evaluate Models Accordingly?, *Journal of Climate*, *31*(13), 4991-5014.

Thompson, V., Dunstone, N.J., Scaife, A.A., Smith, D.M., Slingo, J.M., Brown, S. and Belcher, S.E., 2017. High risk of unprecedented UK rainfall in the current climate. *Nature Communications*, *8*(1), pp.1-6.

2. There are no physical pathways demonstrated for the differences (or similarities) in the modulation of the TEI. For example, what is the exact constructive or destructive pathway that causes AO or ENSO to be a source of predictability? The line "whether the real world behaves the same way…" demonstrates that the authors have not examined this in reanalysis or observed data, which is very concerning.

The reviewer is correct that there is further work to be done to better understand the results seen in this model and to check whether other models and observations do the same thing. In the discussion section we make this point:

"Two key questions are whether the ENSO and AO teleconnections in TEI found here are present in other climate models and in reanalysis and whether relations with TEI translate to relations with severe thunderstorm reports. The latter question has been explored for the ENSO signal (Allen et al., 2015) but not for the AO and not for ENSO and the AO jointly."

We have added the following text to highlight the need for further work and potential extensions.

"As far as we know, this is the first study to examine the constructive/destructive interference of the ENSO and AO signals. Interference of the ENSO and AO signals may also be present in near-surface temperature and precipitation. Regarding the physical mechanisms behind this interaction, one clue might be the fact that the midlatitude jet stream tends to be farther north during both La Nina and positive AO conditions."

We find the prospect of new research avenues to be an exciting aspect of the work here.

3. The authors mention the AO being the dominant mode of hemispheric variability which influences North American near-surface temp. and precipitation. This (as mentioned) is only during the **cold season** when US tornado counts are at an absolute minimum,

during months that the authors do not even examine. It is hard to reconcile this obvious dipole.

The original text is: "The AO is a dominant mode of hemispheric variability which influences North American near-surface temperature and precipitation, especially during the cold season" which seems reasonable to us. ENSO variability is also largest during the cold season. The choice of months analyzed matches the time of year when ENSO and AO influences are waning and the severe weather season is ramping up. The use of the term dipole here is unclear to us: "It is hard to reconcile this obvious dipole."

4. The figures concern me about the TEI index itself and some of the conclusions drawn in the study. First, most paneled spatial plots indicate anomalies of the TEI (which is said to serve as a proxy for tornadic storms) where the background climatology should be zero. I illustrate this by showing the US and Canada tornado reports for the three study months below:

The reviewer has not stated which figures and conclusions are concerning. One guess is that the reviewer is concerned by figures 3 and 4 which show that TEI in CFSv2 is predictable in regions such as the coast and eastern border of British Columbia where the TEI climatology (Figure S1) is quite low and where few if any tornadoes have been reported. On the other hand, we draw no conclusions in the manuscript about such regions. So perhaps this is not the reviewer's concern.

We have made some additional diagnostics of TEI predictability in this region and see no indication that the calculations shown are incorrect. We have added the following text that mentions a bias in TEI for this region and time of the year.

"Both predictability measures show some regions west of 100W and north into British Columbia where TEI in CFSv2 is predictable. On the other hand, few if any tornadoes or thunderstorms are reported in some of these regions, and the findings here may reflect a previously noted positive bias of TEI compared to tornado reports in environments that have relatively high SRH and low cPrcp during this time of the year (Tippett et al., 2014)."

The reviewer has not indicated what report data and period is being shown in their plots. If the Canadian report data shown are freely available, we would appreciate a pointer to it.

5. None of the spatial maps have statistical significance which should also incorporate the false discovery rate.

We have added statistical significance and FDR information to the methods section and to the composite, correlation, and probability maps (Figs. 1, 2, 4, and S5–S8). With the large sample size here from model runs it might be argued that statistical significance might not be the best indicator of robustness. Significance tests for regression maps (Fig. S4) and correlation maps are equivalent (Fig. S5), and we report the significance results in terms of correlation. Significance

tests for composite maps (Fig. 2) and MSESS maps are equivalent (Fig. 3), and we report the significance results in terms of the composites.

6.  TEI, AO, and ENSO, calculations used by the authors from CFSv2 are not available. Thus, the study is not reproducible in its current form.

The reviewer overlooked the link provided in original manuscript to where we downloaded the CFSv2 data  http://iridl.ldeo.columbia.edu/SOURCES/.NOAA/.NCEP/.EMC/.CFSv2/
All CFSv2 monthly variables are available there to download in a variety of formats including binary, netcdf, csv, and opendap. Server-side data subsetting is provided to reduce download size, and download scripting is straightforward. We used only variables provided directly in the CFSv2 output. The formula for TEI is given in the manuscript and is a direct calculation. Nino 3.4 is a box average of SST, and the AO index is the first PC of Z1000.

**Additional comments:**
Throughout the manuscript, it became very noticeable that the authors focused on "self-selected" citations. The first example of this is on line 33 during the discussion of the response of thunderstorm ingredients via radiative forcing. The authors cited a paper by Lepore et al., while omitting numerous other manuscripts that have examined the subject. This happens in (at least) a dozen other places in the manuscript. The last two paragraphs are particularly lazy in this regard.

We disagree with the assessment of laziness and note that suggestions for the inclusion of additional literature are more constructive when accompanied by the relevant citations.

Regarding the last two paragraphs, we carefully selected references of substantial relevance for the scientific argumentation of the manuscript. If there are other references that are relevant to our discussion and that we have overlooked, we are happy to add them.

Specifically, the next to last paragraph in the Discussion makes the following points:
1.  TEI is a product of two variables and our previous work provided a mathematical explanation for why the variance of TEI is higher when its mean is higher. We are not aware of another reference for this point.
2.  The projected climate change signal in TEI is due to an increase in convective precipitation, and we cited our previous work because it computes TEI in CMIP6 models. We have added a reference to CMIP5 results using CAPE and bulk shear.
3.  Observed trends in storm reports have been related to trends in SRH. We have cited the papers that we are aware of that make this point. A number of studies report trends in storm reports and proxies without identifying the environmental factors that are associated with the trend. That omission is relevant because the point being made here is that the projected climate signal and the ENSO/AO signal have distinguishable signatures in the environmental factors. That is why we do not cite here work on trends that does not examine the environmental factors. There may be relevant work that we have overlooked.

4. Our previous work suggests that intensity changes (EF ratings) are more closely tied to changes in SRH, and therefore ENSO/AO variability which has SRH changes would be expressed in relative intensity increases (slope of the intensity/frequency curve) but if the projected climate change signal is primarily in CAPE and not in SRH, there would not be relative intensity changes. We do not know another reference for this point about the sensitivity of intensity to changes in individual environmental factors. There may be relevant work that we have overlooked.
5. We have added two references for the point that the observed trends could be related to internal variability or to a climate change signal that projections get wrong.

As evidence of the diversity of viewpoints, we mention that Reviewer 1 said "these comments are insightful, and should spur additional studies."

The last paragraph points to remaining questions and future work.
1. Checking if the TEI signal is seen in reports (done for ENSO, not for AO). Again, we are happy to add appropriate references.
2. Using the seasonality and teleconnection patterns seen here to enhance signal/noise ratio in future report-based studies. We don't have any references for this approach and only cite references for the AO timescale.

Since the response to radiative forcing is not the topic of the manuscript, an exhaustive set of references in the introduction does not seem appropriate to us. We have cited a recent publication and added "and references therein." We are happy to add other appropriate references.

We would perhaps be able to respond more substantively to the comment about a dozen other places in the manuscript if more details or relevant citations were provided. We are happy to add any appropriate references that we have overlooked.

6. Why use a three-year average of a highly variable quantity? 15+ years should be used here for a climatology.

The 3-year average is provided by SPC and is adequate for its qualitative use here.

7. ..."and not severe thunderstorm activity". There are plenty of papers examining model forecasts of severe thunderstorm activity on the weekly and monthly timescales.

The original text is: "Model-based climate predictability studies have tended to focus on quantities such as near-surface temperature and precipitation, and not severe thunderstorm activity, in large part because climate models do not resolve thunderstorms." We stand by our statement that there are relative few predictability studies (as opposed to case studies or forecast assessments) for severe thunderstorms but would be happy to add references that we have overlooked.   We would perhaps be able to respond more substantively to this comment if it were accompanied by the relevant citations.

8. The discussion re: sources of predictability on the non-ENSO timescale lacks context regarding many other works that have examined such timescales. One example of this would be the growing body of literature examining the MJO impacts on thunderstorms. These should be discussed here.

We have not attempted or claimed to make an exhaustive survey of the sources of thunderstorm predictability in the Discussion.  We have mentioned the projected climate change signal in the Discussion because it differs from the ENSO and AO ones that we see here and because that difference might be useful in future work. We do not see a direct connection between the monthly analysis here and MJO signals which do not persist that long. In particular, the monthly data here could not credibly be used to analyze the MJO signal. If there is a connection that we have overlooked we would be happy to hear it and add an appropriate reference.

9. The authors mention the AO being the dominant mode of hemispheric variability which influences North American near-surface temp. and precipitation. This is only during the cold season when US tornado counts are at an absolute minimum. It is hard to reconcile this obvious dipole.

See our response to comment 3.

10. Line 72. What modification of additional variables relative to the previous study?

The earlier study did not use Z1000 or grid point values of TEI. Those are the additional variables.

---

## Author Response (AR2)

Black second set of reviewer comments
Blue previous responses
Green current responses

Responses to author responses on major comments:

1) At the very least, the title should add in "Model Predictability..."

The abstract is clear that the analysis is "in climate model forecasts," and predictability studies are often based on model behavior. So, we think the title is appropriate.

We do not find the reviewer's argument for adding "model" to the title entirely compelling. If editor sees a problem with the title, we are willing to change the title.

2) The authors did not address this concern. They should examine the relationship of the physical pathway and/or composited synoptic weather features to understand this modulation and pathway. Without it, there is no physical basis for the predictability relationships and it would be rather useless from an operational perspective.

We tried to address this point in our responses to the first review and made changes in the text that we thought were appropriate. It is unclear what parts of our response were inadequate. It is unclear to us what specific analysis the reviewer is suggesting now: "They should examine the relationship of the physical pathway and/or composited synoptic weather features to understand this modulation and pathway." Also the last point about the "operational perspective" is unclear to us since we make no claims that our finding are ready for use in operational forecasting. We think that our composites analyses are sufficient to support our findings within the context of physics-based climate models which is the setting of our analysis.

3) OK, but AO is not the dominant mode of variability for severe weather for the months examined in this study. The MJO is a much larger factor and is not addressed.

We are glad that the reviewer accepts our response. We do not claim anywhere that the AO is the dominant mode of variability for severe weather in the months considered in this study.

As we noted in our first response, we do not see a direct connection between the monthly analysis here and MJO signals which do not persist that long. In particular, the monthly data here could not credibly be used to analyze the MJO signal. If there is a connection that we have overlooked we would be happy to hear it and add an appropriate reference.

4) The author responses to comment 4 are adequate. A google search points directly to the Canadian tornado data: https://open.canada.ca/data/en/dataset/65658050-7a80-4da3-9a09-da137c203a34

5) The author responses to comment 5 are adequate.

6) The response re: TEI data is not adequate for an open source journal in my opinion. Yes, the CFSv2 data is available via the URL, but the calculated TEI data are not. The TEI, AO, and ENSO derived data should be provided in a repository to ensure reproducibility.

We used only variables provided directly in the CFSv2 output. The formula for TEI is given in the manuscript and is a direct calculation. Nino 3.4 is a box average of SST, and the AO index is the first PC of Z1000.

As the reviewer notes, the CFSv2 data is available. TEI is directly computed from that CFSv2 data in one line using the formula given in the manuscript.

TEI = exp(-14.01 + 1.36*log(cPrcp) + 1.89*log(abs(SRH))) x days per month

Addl. comments:

The others mention, "We are happy to add other appropriate references." My original suggestion was that the paper initially read like a literature review of only the authors' works. Adding relevant citations doesn't cost anything and would enhance the breadth of the manuscript for readers that are unfamiliar with these areas of research.

… suggestions for the inclusion of additional literature are more constructive when accompanied by the relevant citations.

In the first response, we gave a detailed and lengthy explanation for the choice of citations in the Discussion and repeatedly asked for additional references that we might have overlooked. We don't understand why the reviewer questions the choice of citations but does not suggest additional or alternative citations. The WCD obligations for referees state that "Any statement that an observation, derivation, or argument had been previously reported should be accompanied by the relevant citation."

I disagree with the comment about a three-year average. 15+ years should be used in a climatological context.

Our original submission included preliminary report numbers for 2021, which were the data then available. Actual report numbers were provided by NOAA/SPC on 11 July 2022 for 2021. We are now able to follow the reviewer's suggestion and compare 2021 numbers with those of the period 2006–2020, and we have revised the text accordingly.